# A Recursive Approach for Magnetic Field Estimation in Spacecraft Magnetic Attitude Control

Mohammed A. A. Desouky [1,*,†] and Ossama Abdelkhalik [2]

[1] Department of Mechanical Engineering, Michigan Technological University, Houghton, MI 49931, USA,
[2] Department of Aerospace Engineering, Iowa State University, Ames, IA 50011, USA
* Correspondence: madesouk@mtu.edu
† Current address: Egyptian Armed Forces, Cairo, Egypt.

**Abstract:** This paper is concerned with magnetic attitude control of spacecraft. The operation of the magnetic actuators is usually on a duty cycle; during the off times in this duty cycle the magnetometers are used to measure the magnetic field around the spacecraft. This alternate operation of magnetic actuators and sensors avoids the noise effect on the magnetometers coming from the magnetic actuators. This alternate operation results in longer maneuver times. This paper presents an estimation approach for the magnetic field, as well as the spacecraft attitude, that increases the duty cycle of the magnetic rods while reducing the rate of collecting the magnetometer data. A modified Multiplicative Extended Kalman Filter (MEKF) is used in the proposed approach. A relatively simple and fast dynamic model is developed for use in the MEKF. Monte Carlo simulations presented in this paper show that the proposed approach results in less maneuver time, and less power consumption by the magnetic rods when compared to a standard magnetic control approach. The magnetic field estimation process is verified using data collected from the CASSIOPE spacecraft using its telemetry system and the results are presented.

**Keywords:** magnetic attitude control; attitude estimation; magnetic field estimation; magnetic rods' duty cycle

## 1. Introduction

The Attitude Determination and Control System (ADCS) plays an essential role in all spacecraft missions. A high-performance spacecraft ADCS is usually more expensive, such as those that use star trackers, gyroscopes, and momentum exchange devices as attitude actuators. Small low-cost spacecraft, on the other hand, are being considered for several types of missions. Because of their dependability, low cost, lightweight, and energy efficiency, these spacecraft typically employ low-cost components such as magnetic rods and magnetometers [1–3]. A worldwide survey of small satellite missions revealed that the foremost commonly used sensors are sun sensors and magnetometers. Furthermore, about 40% of the nanosatellites have magnetic rods for active magnetic attitude control [4]. There are, however, some challenges. The magnetic rods usually suffer from poor accuracy and instantaneous under actuation [5,6]. The generated torque is constrained to be in the plane that is orthogonal to the ambient magnetic field vector. Therefore, a three-axis magnetic attitude control is merely possible if the orbit sees a magnetic field variation that is sufficient to ensure the spacecraft stability, which is a requirement that is usually fulfilled in inclined orbits [1]. Furthermore, as mentioned in [7], there are additional restriction criteria on the spacecraft inertia tensor to ensure controllability.

Measuring the spacecraft's external magnetic field using magnetometers serves two purposes. The first is to use magnetic field data to compute the magnetic dipole moment (attitude control) and the second purpose is to estimate the spacecraft attitude and rate. There are several approaches that can be used for the latter purpose. Psiaki et al. [8] proposed an Extended Kalman Filter (EKF) for attitude, rate, and constant

disturbance torque estimation based on magnetic field measurements and their time derivatives. Tortora et al. [9] proposed a fast angular rate estimation scheme using magnetometer readings, assuming that the inertial ambient magnetic field vector does not significantly change during the short sampling time. An analytic approach is used in [9] that does not require attitude information. Humphreys et al. [10] developed a spinning spacecraft with wire booms, a filter, and smoother based on magnetometer information for estimating the attitude, rate, and boom orientations. Abdelrahman and Park employed magnetometer measurements and their time derivatives with Sigma-Point Kalman Filter for spacecraft three-axis attitude control and rate estimation. This filter's capability in estimating the attitude is better than 5 deg, and the rate error is on the order of 0.03 deg/s in each axis [11]. In the above-cited studies, the magnetometer measurements are compared with the geo-magnetic field information from a high-order Earth magnetic field model, such as the World Magnetic Model (WMM) or the International Geomagnetic Reference Field (IGRF) or (T89) model. It is worth noting that these models are used for calibrating magnetometers also as in [12–15]. Based on magnetometer and Sun sensor information, Soken and Sakai [16] developed an inertial vector attitude estimate approach for small spacecraft. The technique starts with a basic attitude determination using the TRIAD approach, then introduces the estimated attitude to an Unscented Kalman filter for accurate estimation via magnetometer calibration in real-time. Altuntas et al. [17] proposed a cascaded filtering scheme in which the QUEST method was employed to update the Multiplicative Extended Kalman Filter at low angular rates with just magnetometer readings. Ivanov et al. [18] developed, uploaded to SiriusSat-1, and analyzed an extended Kalman Filter in real time to follow the magnetometer bias induced by the onboard magnetic dipoles in order to improve the accuracy of attitude motion estimates. The attitude of a spacecraft spinning at about orbital speed is measured with 3–4° accuracy, with magnetometer bias estimation accuracy in the 400 nTesla range [18]. Using magnetometer and sun vector readings, Pourtakdoust et al. [19] constructed a modified Square Root unscented Kalman filter with bounded noise characteristics for attitude and angular rate estimation. Furthermore, the authors proposed a strategy for optimizing the installed sensor orientation in order to decouple the residual signal, which impacts attitude estimate accuracy [19]. To that purpose, several recent research studies, such as [20–22], have employed magnetometer readings to determine attitude.

Other different sensors for low-cost missions, such as the Sun sensor, Earth horizon and even the gyroscope are utilized for attitude estimation as in [23,24]. There is a variety of attitude estimation techniques that can be found in the survey of nonlinear attitude filtering methods in [25], and also in the review on the quaternion-based methods for spacecraft attitude determination in [26].

Magnetometer measurements are also used in feedback control in several algorithms of spacecraft attitude stabilization, and detumbling maneuver, as in [27–42] and in the survey papers by Silani and Lovera [1], and by Ovchinnikov and Roldugin [43]. In these algorithms, the control analysis and design assume continuous-time magnetometer measurements and continuous actuation of the magnetic rods. The fact is magnetometer measurements are often degraded by the time-varying magnetic field generated by the currents in magnetic rods and other spacecraft electronics. Furthermore, the measurements themselves are prone to several types of errors; these errors include null-shift error, the non-linearity error, non-orthogonality error, the sensor noise, hard-iron error and soft-iron errors, in addition to the static and time-varying biases [12,13].

One mitigation technique is to use a boom to have a physical separation between the magnetometer and the other spacecraft electronics [44,45]. The boom structure, however, results in more complexity for the whole spacecraft system. Another mitigation technique that is usually performed in small low-cost spacecraft missions is to operate the magnetic rods and the magnetometers at alternate times, to eliminate the effect of the rods on the magnetometers [2,46,47]. Celani [46] presented a magnetic state feedback attitude control law, taking into account the intermittent activation of the magnetic rods and magnetometers during the operation, where a systematic method is proposed for the choice of the sampling

period of the updated command and in which, the generated dipole moment is on the basis of piece wise constant command. Celani [46] assumed that the period that is dedicated to the magnetometer measurements is small enough to be ignored. Celani [47] extended his work by considering the magnetometer measurement period during the design phase of the control law. A design method and systematic approach are obtained for the selection of the controller's parameters and actuation interval length, respectively [47]. The design analysis of magnetic attitude control systems usually neglects this intermittent operation of the magnetic rods, leading to under-estimation of the required maneuver time and the rod's power consumption [2]. Desouky and Abdelkhalik [2] developed an algorithm featured by its low frequency in collecting magnetometer measurements with a higher magnetic rod duty cycle ratio. This algorithm resulted in improvements in the required maneuver time and magnetic rod power consumption at the expense of the required computational demands.

Motivated by the above challenges, this paper presents an attitude estimation and control algorithm that improves the maneuver time and power consumption that is required by magnetic rods. This algorithm estimates the spacecraft attitude, the spacecraft angular rate, and the magnetic field using a modified version of the Multiplicative Extended Kalman Filter (MEKF). The proposed approach uses a magnetometer, gyroscope, and a measurement for an inertial vector such as the direction of the Sun (Sun sensor). For three-axis attitude control, it is assumed that only magnetic actuators are used. The proposed concept is to eliminate the need to turn off the magnetic rods, in some cycles, to increase the rod's operation time. This is achieved by using the estimated magnetic field in these cycles instead of the measured one. The key concept of estimating the magnetic field starts by computing a pseudo magnetic field measurement, leveraging the existing spacecraft's angular velocity feedback loop to probe the magnetic field. This pseudo-measurement is further refined inside the MEKF using a proposed simple magnetic field model. It is shown in this paper that the proposed algorithm can achieve improvements in the power consumption and the maneuver time. The proposed work is assessed via Monte Carlo simulations. The magnetic field estimation is validated using data from the CASSIOPE spacecraft obtained through its telemetry system.

This paper is an extension of the work in [3] and different than the work in [2] in the following aspects: (1) computing the pseudo measurements of the magnetic field by measuring the spacecraft response to a known control command is implemented here from a geometric point of view, as opposed to the Tikhonov regularization technique used in [2], (2) this work uses an attitude sensor, in addition to the magnetometers, in the MEKF, as opposed to using only magnetometers measurements in [2], and (3) the computational load of the proposed work here is much less than that in [2], and is comparable to the computational load of most exciting magnetic control techniques within the literature, as are going to be shown within the simulation results. This paper is organized as follows. Section 2 briefs the spacecraft models. The proposed ADCS is presented in Section 3. Section 4 presents the Monte Carlo simulation results. A validation process using real telemetry data with detailed procedures to verify the proposed algorithm is presented in Section 5.

## 2. Spacecraft Dynamic Model

This part contains standard information that is used to complete the presentation. Here the coordinate reference frames are presented, as well as the kinetic and kinematic models. For the attitude determination and control algorithm, the following reference frames are used:

1. Earth-Centered Inertial frame (ECI). The Earth's center is the ECI frame origin. This reference frame is denoted *i*, and the earth rotates around its Z-axis. The X-axis points towards the vernal equinox, where the Y-axis complies with the right-handed triad system.

2.　Satellite body frame: This frame's origin is at the satellite's centre of mass. The axes are chosen to align with the central principal axes of the spacecraft. The body frame is denoted $b$.

In the remainder of this paper, [.] represents a matrix, bold symbols represent vector such as $\mathbf{A}$, $[\mathbf{A}]_x$ is a skew-symmetric matrix whose elements are the elements of the vector $\mathbf{A}$ that represents the cross product of $\mathbf{A} \times \mathbf{B} = [\mathbf{A}]_x\mathbf{B}$, where $\mathbf{B}$ is a vector, $\tilde{\mathbf{B}}$ represents an estimation of the vector $\mathbf{B}$, $\delta\mathbf{B}$ represents an error vector added to vector $\mathbf{B}$ to obtain the estimated vector $\tilde{\mathbf{B}}$, as $\tilde{\mathbf{B}} = \mathbf{B} + \delta\mathbf{B}$, $\hat{\mathbf{C}}$ represents a unit vector in the direction of the vector $\mathbf{C}$, and $\bar{\mathbf{D}}$ represents the linearization point of the vector function $\mathbf{D}$ (the linearizion process is carried out at $\bar{\mathbf{D}}$). The $^i\mathbf{v}$ represents vector $\mathbf{v}$ defined in the inertial frame $i$. For notation simplification, any vector without a pre-superscript is defined in the ($b$) frame. The $\boldsymbol{\omega}_{bi}$ represents the angular velocity of the ($b$) frame with respect to the ($i$) frame , expressed in the ($b$) frame.

The spacecraft attitude in this study is represented by the quaternion to avoid singularity. The attitude kinematics can be written as:

$$\dot{\mathbf{q}}_{bi} = \begin{bmatrix} \dot{q}_{0(bi)} \\ \dot{\mathbf{q}}_{v(bi)} \end{bmatrix} = \frac{1}{2}\begin{bmatrix} -\mathbf{q}_{v(bi)}^T \\ q_{0(bi)}[\mathbf{1}_{3x3}] + [\mathbf{q}_{v(bi)}]_x \end{bmatrix}\boldsymbol{\omega}_{bi}, \tag{1}$$

where $\boldsymbol{\omega}_{bi} \in \mathbb{R}^3$ is the spacecraft angular velocity of the body frame with respect to the inertial frame, expressed in the body frame. The vector $\mathbf{q}_{bi} \in \mathbb{R}^4$ is the quaternion, and $\mathbf{q}_{bi} = \begin{bmatrix} q_{0(bi)} & \mathbf{q}_{v(bi)}^T \end{bmatrix}^T$. Let $\mathbf{q}_{v(bi)} = \begin{bmatrix} q_{1(bi)} & q_{2(bi)} & q_{3(bi)} \end{bmatrix}^T$. The matrix $[\mathbf{1}_{3x3}] \in \mathbb{R}^{3\times3}$ is unity matrix. The transformation matrix $[R(\mathbf{q})]_{bi}$ from the $i$ to the $b$ frame is computed using the quaternion as follows:

$$\begin{aligned} [R(\mathbf{q})]_{bi} &= (q_{0(bi)}^2 - \mathbf{q}_{v(bi)}^T\mathbf{q}_{v(bi)})[\mathbf{1}_{3x3}] \\ &+ 2\mathbf{q}_{v(bi)}\mathbf{q}_{v(bi)}^T - 2q_{0(bi)}[\mathbf{q}_{v(bi)}]_x \end{aligned} \tag{2}$$

The attitude dynamics of a rigid spacecraft are expressed using Euler's equations as follows [48,49]:

$$[I]\,\dot{\boldsymbol{\omega}}_{bi} = -\boldsymbol{\omega}_{bi} \times [I]\,\boldsymbol{\omega}_{bi} + \mathbf{T} + \mathbf{T}_d, \tag{3}$$

where $[I] \in \mathbb{R}^{3\times3}$ denotes the spacecraft inertia matrix. As previously stated, the inertia matrix $[I]$ is fixed in the body frame and is a diagonal matrix since the body frame is aligned with the principal axes. The disturbance torque $\mathbf{T}_d$ here represents the sum of the solar radiation torque $\mathbf{T}_{sr} \in \mathbb{R}^3$, the aerodynamic torque $\mathbf{T}_{aero} \in \mathbb{R}^3$, the gravity gradient torque $\mathbf{T}_{gg} \in \mathbb{R}^3$, the residual magnetic torque $\mathbf{T}_{rsd} \in \mathbb{R}^3$ due to the residual magnetic field generated by spacecraft electronics including the magnetic These torque models are summarised in [28,42]. The control torque on the spacecraft, $\mathbf{T} \in \mathbb{R}^3$, is here assumed to be due to only the three magnetic coils, and hence it is not feasible to create a torque along the magnetic field vector, since

$$\mathbf{T} = \mathbf{M} \times \mathbf{B}, \tag{4}$$

where $\mathbf{B} \in \mathbb{R}^3$ and $\mathbf{M} \in \mathbb{R}^3$ represent the ambient magnetic field vector and dipole moment vector, respectively. A detailed description of The dipole moment model as a function of the current that is generated by the magnetic rods is presented in [28,42].

## 3. Attitude Determination and Control System

This section presents the proposed algorithm for spacecraft attitude stabilization and control. First, a reference ADCS algorithm is discussed highlighting the challenges that will be addressed in the proposed algorithm. This reference ADCS algorithm is utilized for comparison and performance evaluation of the proposed ADCS.

In the reference ADCS algorithm, the magnetic rods and the magnetometers are assumed to operate at alternate times, to avoid high noise on the magnetometers' measure-

ments during the operation of the magnetic rods. This alternate operation means that the magnetic rods operate with a certain duty cycle, as shown in Figure 1. The sampling time $T_s = t_{i+1} - t_i$ (sometimes referred to as the cycle period) depends on the rate of update in the control system.

Let $T_{dc}$ be the magnetic rods duty cycle which is the maximum time period in which the magnetic rods can be turned on, in one cycle period, and let $\delta \leq 1$ be the duty cycle percentage or ratio. Hence, we can write: $T_{dc} = \delta T_s$. Usually, each cycle period includes also a period for magnetometer measurements, a period for raising the magnetic rods currents from zero to the required value ($\mathbf{i}_c$), and a period for magnetic rods desaturation dedicated to reducing the generated field in the rods. The fall period, in which the current drops to zero, is part of the desaturation period. The lower part of Figure 1 shows several cycles of the magnetic rods and the magnetometer activation periods. The attitude estimation algorithm typically updates the estimates for the quaternion $\tilde{\mathbf{q}}$, the spacecraft angular velocity $\tilde{\omega}$, and/or gyroscope bias $\beta$ each cycle. The upper part of Figure 1 shows the estimated and measured quantities, and the times at which they are collected. The measurements are: the angular velocity $\omega$, the magnetic field $\mathbf{B}$, and the sun direction $\mathbf{V}_{sun}$.

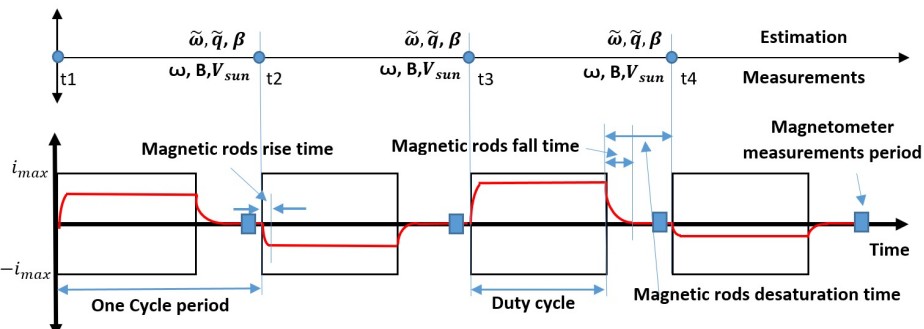

**Figure 1.** In the reference ADCS algorithm, the magnetic rods operate on a duty cycle principal, alternating with magnetometers to avoid high noise on the latter.

Control algorithms that do not account for the above magnetic rod's duty cycle usually underestimate the maneuver time and the rod's power consumption [2]. Increasing the duty cycle ratio $\delta$ would reduce the actual maneuver time, and enhance the steady-state error and system stability [6].

In this study, the proposed ADCS algorithm increases the activation time of the magnetic rods, i.e., the duty cycle ratio. This is achieved by estimating the magnetic field parameters at some of the cycles and hence eliminating the need for magnetic field measurements in these cycles. The lower part of Figure 2 illustrates a scenario where the magnetic field is measured every three cycles while counting on the estimated magnetic field in the cycles that do not have magnetometer measurements at times $t_2$ and $t_3$. At the core of the proposed ADCS is an algorithm that estimates the magnetic field in these cycles (at times $t_2$ and $t_3$) when the magnetometer measurements are not available; this algorithm is presented in Section 3.2. The upper part of Figure 2 illustrates the measured and estimated quantities in each cycle. At each of the times, $t_2$ and $t_3$, a calculated magnetic field $\mathbf{B}_{sdo}$ is used as a pseudo measurement input to the estimator to estimate the magnetic field $\tilde{\mathbf{B}}$ that is used for control. While, at times $t_1$ and $t_4$, the direct true magnetic information measurements $\mathbf{B}$ are used for control and attitude estimation.

Hence, a longer duty cycle for the magnetic rods becomes possible. The equivalent duty cycle ratio $\delta_{eqv}$ for the proposed ADCS can be computed as follows:

$$\delta_{eqv} = 1 - \frac{1 - \delta}{\epsilon} \tag{5}$$

where $\epsilon$ is the number of cycle periods between the real magnetometer measurements in the proposed algorithm.

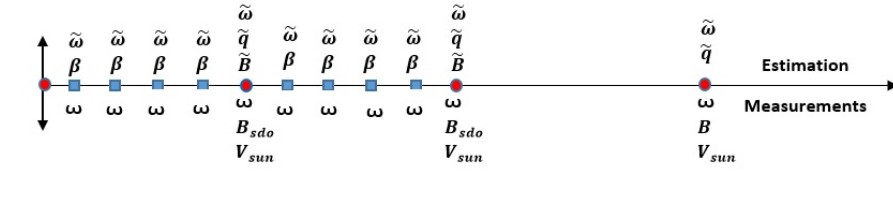

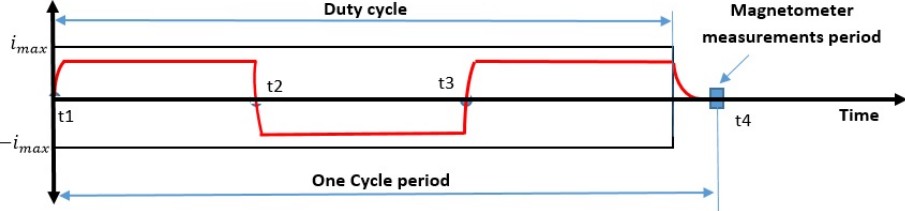

**Figure 2.** The proposed magnetic rod operation.

The lower section of Figure 2 depicts a situation in which $\epsilon = 3$. Consider the following standard (reference) control algorithm: $\delta = 0.7$ and $T_s = 0.1$ [s]. The proposed control algorithm's duty cycle would be $\delta_{eqv} = 0.9$ for $\epsilon = 3$. Let $f_c$ be the update frequency in the control system. For the sake of comparison, $f_c$ is assumed constant in this research at the rate $f_c = 1/T_s$, where $T_s$ is the sampling time of the reference ADCS. Section 3.1 presents the control law. Section 3.2 presents the magnetic field pseudo measurement $\mathbf{B}_{sdo}$ computation algorithm. The modified Multiplicative Extended Kalman Filter used in the proposed ADCS is presented in Section 3.3.

### 3.1. Control Law

In this paper, the PD-like control logic presented in references [41,50,51] is adopted. The process of computing the control torque starts with computing a control term $\mathbf{T}_{req}$, for inertial pointing maneuvers as follows:

$$\mathbf{T}_{req} = -(\zeta^2 k_p \mathbf{q}_{v(err)bi} + \zeta k_d \delta \boldsymbol{\omega}_{bi}),\tag{6}$$

where $\mathbf{q}_{v(err)}$ is the quaternion error vector part between the desired attitude and the current one, $\delta \boldsymbol{\omega}_{bi}$ is the spacecraft angular velocity error between the desired rate and the current one, $k_p > 0$ is the proportional gain, $k_d > 0$ is the derivative gain, and $\zeta$ is a parameter introduced to limit the controller gains due to the time-varying nature of the ambient magnetic field and consequently limit the settling time of the attitude orientation. The control limit parameter is bounded: $0 < \zeta < \zeta^*$. This control ensures that the equilibrium point is locally exponentially stable [41,50]. Due to the singularity caused by the cross product in Equation (4) [52], the required dipole moment cannot be determined from a given command control torque. Furthermore, the resultant torque is limited to acting directly in the plane perpendicular to the magnetic field vector for a given dipole moment. To solve this issue, consider $\mathbf{T}_{req}$ to be the desired torque vector, and $\mathbf{T}$ to be the torque vector equal to the projection of $\mathbf{T}_{req}$ on the plane perpendicular to the ambient magnetic field vector $\mathbf{B}$. The vector $\mathbf{T}$ is located in the plane in such a way that the Euclidean norm of the residual vector (or the angle between the vectors $\mathbf{T}_{req}$ and $\mathbf{T}$) is reduced [1,34]. Using the estimated magnetic field vector $\tilde{\mathbf{B}}$, the required dipole moment, at times $t_2$ and $t_3$ in Figure 2, is calculated as follows:

$$\mathbf{M} = \frac{\tilde{\mathbf{B}} \times \mathbf{T}_{req}}{\|\tilde{\mathbf{B}}\|^2}\tag{7}$$

Combining Equations (4) and (7), the applied torque becomes:

$$\mathbf{T} = -\mathbf{B} \times \frac{\tilde{\mathbf{B}} \times \mathbf{T}_{req}}{\|\tilde{\mathbf{B}}\|^2} = [\Gamma(t)] \frac{\mathbf{T}_{req}}{\|\tilde{\mathbf{B}}\|^2}, \tag{8}$$

where the matrix $[\Gamma(t)] = [\mathbf{B}]_x [\tilde{\mathbf{B}}]_x^T$ is semi-positive definite, when $\tilde{\mathbf{B}} \approx \mathbf{B}$. In this case, the estimated magnetic field $\tilde{\mathbf{B}}$ is used for computing the required dipole moment, at times $t_2$ and $t_3$ in Figure 2.

As demonstrated in reference [53], this time-varying non-autonomous system is controllable over an indefinite time. The stability of such a system is discussed in various references, including [54], where the generalized average theory was used to demonstrate that the trajectories of non-autonomous systems remain near to the trajectories of its average if $\zeta < \zeta^*$. As a result, the non-autonomous term $[\Gamma(t)]$ in this issue may be substituted by its average, $[\Gamma_{av}]$, which is defined as follows [50]:

$$[\Gamma_{av}] = \lim_{t \to \infty} \frac{1}{t} \int_0^t [\mathbf{B}]_x [\tilde{\mathbf{B}}]_x^T dt \tag{9}$$

References [50,53] show that when $\tilde{\mathbf{B}} = \mathbf{B}$, the average matrix $[\Gamma_{av}]$ is positive definite. Further, if we assume that the equilibrium is at $\delta\boldsymbol{\omega}_{bi} = 0$ and $\mathbf{q}_{err} = [1\ 0\ 0\ 0]^T$, then reference [55] shows that, as time goes to infinity, the control law guarantees $\delta\boldsymbol{\omega}_{bi}$ and $\mathbf{q}_{err}$ go to equilibrium. Moreover, it was shown in references [41,50,51] that this control law guarantees local asymptomatic stability at the equilibrium point, given the positive definiteness of $[\Gamma_{av}]$. References [55,56] as well prove this asymptomatic stability using a Lyapunov approach, assuming that $\zeta$ is bounded by $\zeta^*$, and that $\zeta^*$ is a decreasing function of $k_p$ and $k_d$ [57]. The above-mentioned analysis about the control law (6) assumed that the magnetic rods are operated in a continuous mode by ignoring the duty cycle effect. Desouky and Abdelkhalik [6] presented analytically that increasing the duty cycle ratio will reduce the attitude steady-state error. Which can be considered another benefit of the proposed work here.

### 3.2. Computation of Magnetic Field Pseudo Measurement

The following discussion is dedicated to finding the best guess of the ambient magnetic field vector around the spacecraft when the magnetometer measurements are not available. This best guess, $\mathbf{B}_{sdo}$, is used as a pseudo measurement. Knowing the kinetic model of the spacecraft (Equation (3)) it is possible to use the angular velocity measurements to obtain an estimate for the spacecraft applied torque, $\tilde{\mathbf{T}}$. This torque $\tilde{\mathbf{T}}$ is then used to compute the pseudo measurement of the magnetic field $\mathbf{B}_{sdo}$.

The numerical evaluation of $\dot{\boldsymbol{\omega}}$ has a significant impact on the accuracy of the obtained results in this approach. Hence, the five-point stencil method [58] is used to evaluate the time rate of change of $\boldsymbol{\omega}$. The formula for computing $\dot{\boldsymbol{\omega}}$ is:

$$\dot{\boldsymbol{\omega}}_{k-3h} = \frac{-\boldsymbol{\omega}_k + 8\boldsymbol{\omega}_{k-h} - 8\boldsymbol{\omega}_{k-3h} + \boldsymbol{\omega}_{k-4h}}{12h} + \frac{h^4}{30} \boldsymbol{\omega}_{k-3h}^5, \tag{10}$$

where $h$ is the time step of the angular velocity measurement. It is worth noting that increasing the number of points for the stencil method leads to more accuracy in estimating the applied torque at the expense of the computational cost. According to Abramowitz [58], the inaccuracy of the five-point stencil approach is of order $h^4$ compared to $h^2$ for the two-point difference method. It is assumed in this study that five gyroscope readings are taken within $T_s$, see the upper part of Figure 2. It is also assumed that an average torque value is constant within each $T_s$ period. It is noted that this assumption of having a frequency of gyroscope measurements collection at least five times higher than the control command frequency, is a reasonable assumption, as discussed in [24].

Gyroscope drift rate bias, random walk, scaling factor error, non-linearity error, and misalignment error can all affect gyroscope measurements. In this study, all of the

aforementioned errors—aside from the drift rate bias—are treated as a single normal distribution error with a zero mean. This is a reasonable assumption given that the manufacturer often provides the gyroscope's noise characteristics. On the other hand, gathering several gyroscope observations over an extended period of time is one method for determining the drift rate bias vector. These observations can be compared to those from other sensors, such as star tracker readings, that reveal information about the angular velocities of the spacecraft, where the bias vector can be estimated using an estimation method, such as the least squares method. In accordance with the manufacturer's recommendations, such as once a month, this procedure can be repeated to update the drifting bias.

However, numerous estimating approaches can estimate the bias in cases where another sensor that provides data on the angular velocities of the spacecraft is unavailable, as is the situation in this work. Such as the Kalman filter, where the manufacturer or customer laboratory studies provide the noise characteristic of the gyroscope inaccuracies and drift bias. In this work, whenever the gyroscope measurements are available, an $EKF_\omega$ is employed to estimate the spacecraft's angular velocity and bias vector. Between times $t1 - t2$ and $t2 - t3$, this $EKF_\omega$ outputs the estimated angular velocity that is needed to compute the $\dot{\omega}$ that is shown in the upper portion of Figure 2. It is important to remember that updating the drift rate bias onboard can be implemented periodically to lessen the computing load once an adequate assessment of the drift rate bias has been calculated. However, in this study, evaluating the bias vector will be completed at each sampling time in an effort to provide an idea of the worst-case scenario, in terms of computation load, regarding the use of the proposed work

It is worth noting that the five-stencil approach effectively reduces the drift bias error's impact on the computation of the $\dot{\omega}$. For instance, failing to account for drift rate bias $0.5\,°/\mathrm{hr}$ may result in an error when computing $\dot{\omega}$ equal $(2.68 \times 10^{-32})h^4$ [rad/s$^2$].

The derivative of the angular velocity $\dot{\omega}$ can be computed using Equation (10). Then, an estimation of the torque $\tilde{\mathbf{T}}$ can be computed using the Euler dynamic model Equation (3). Equations (7) and (8) imply that the torque, dipole moment and magnetic field vectors are orthogonal to each other in the ideal case when $\tilde{\mathbf{B}} = \mathbf{B}$. Assuming that $\tilde{\mathbf{B}}$ remains close to $\mathbf{B}$, and given $\mathbf{M}$ and $\tilde{\mathbf{T}}$, it is possible to compute the pseudo measurement vector, $\mathbf{B}_{sdo}$, from a geometric point of view. The unit vector of the $\mathbf{B}_{sdo}$ vector can be computed as follows:

$$\hat{\mathbf{B}}_{sdo} = \frac{\tilde{\mathbf{T}}}{\|\tilde{\mathbf{T}}\|} \times \frac{\mathbf{M}}{\|\mathbf{M}\|} = \hat{\tilde{\mathbf{T}}} \times \hat{\mathbf{M}} \tag{11}$$

In addition, the magnitude of the pseudo measurement of the ambient magnetic field vector can be computed as follows:

$$\|\mathbf{B}_{sdo}\| = \frac{\|\tilde{\mathbf{T}}\|}{\|\mathbf{M}\|} \tag{12}$$

Therefore, the pseudo measurements of the ambient magnetic field vector are:

$$\mathbf{B}_{sdo} = \frac{\|\tilde{\mathbf{T}}\|}{\|\mathbf{M}\|} \hat{\tilde{\mathbf{T}}} \times \hat{\mathbf{M}} = \frac{\tilde{\mathbf{T}} \times \mathbf{M}}{\|\mathbf{M}\|^2} \tag{13}$$

Further analysis over the computation of $\mathbf{B}_{sdo}$ is carried out to check the effect of the error in the torque and the dipole moment vectors, $\delta\mathbf{T}$ and $\delta\mathbf{M}$, respectively. The torque error is due to the noises in the gyro measurements and the errors in modeling the external disturbance torques such as gravity gradient, residual dipole moment, and aerodynamic drag, in addition to the sensitivity of the spacecraft dynamic model to the uncertainty in the moment of inertia. The error in the dipole moment vector $\delta\mathbf{M}$ is due to the uncertainty in the actuator's model and the noises and digitization process in the measured commanded current to the actuators. The estimate of the torque applied on the spacecraft is expressed as:

$$\tilde{\mathbf{T}} = \tilde{\mathbf{M}} \times \mathbf{B}_{sdo} \tag{14}$$

Therefore, the torque error vector is as follows:

$$\delta\mathbf{T} = \mathbf{M} \times \delta\mathbf{B} + \delta\mathbf{M} \times \mathbf{B} + \delta\mathbf{M} \times \delta\mathbf{B} \tag{15}$$

The most dominant term in Equation (15) on the right-hand side is the first term. Therefore, Equation (15) can be approximated as follows:

$$\delta\mathbf{T} \approx \mathbf{M} \times \delta\mathbf{B} \tag{16}$$

where a bound $\gamma \in \mathbb{R}^3$ on the ambient magnetic field error vector, $|\delta\mathbf{B}| \le \gamma$, is added. To make this analysis easier to visualize, we express the error in each vector in terms of the corresponding angle. The angle $\lambda$ is the angle between the computed torque $\tilde{\mathbf{T}}$ and the true torque $\mathbf{T}$. While the angle $\eta$ is the angle between the estimated dipole moment vector $\tilde{\mathbf{M}}$ and the true dipole moment vector $\mathbf{M}$, see Figure 3. Considering Figure 3, it is possible to express the error vector in the ambient magnetic field, $\delta\mathbf{B}$ in terms of the angle $\alpha$ as follows:

$$|\alpha| < \alpha_\gamma, \tag{17}$$

where the angle $\alpha$ between the optimal $\mathbf{B}_{sdo}$ vector and the true one $\mathbf{B}$ should be kept under a threshold angle $\alpha_\gamma$. Figure 4 shows the typical relation between the angles $\alpha$ and $\eta$ for different values of $\lambda$ where $\eta$ is presented on the right vertical axis. For a wide range of the angle $\eta$, there is almost no change in the angle $\alpha$. In the same figure, the relation between the angles $\alpha$ and $\lambda$ for different values of $\eta$ is plotted, where $\lambda$ is presented on the left axis. The correlation between $\alpha$ and $\lambda$ is strong; consequentially $\lambda$ has a significant impact on $\delta\mathbf{B}$, in compliance with Equation (16).

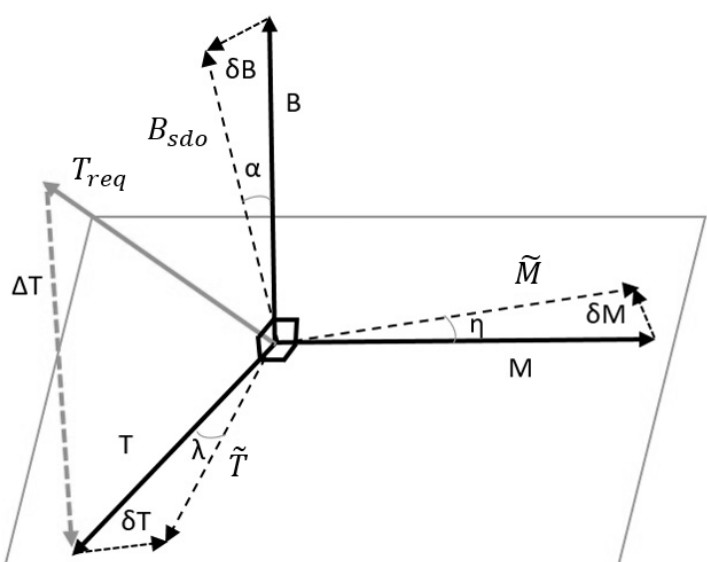

**Figure 3.** The desired $\mathbf{T}_{req}$ and the projected $\mathbf{T}$ torques for a given $\mathbf{B}$ vector.

The angle $\lambda$ can be used as a measure for the accuracy of computing the estimated torque $\tilde{\mathbf{T}}$. Neglecting $\mathbf{T}_d$, Figure 5 shows the angle $\lambda$ history for a test case scenario. In that test case, there is an agreement between the computed torque and the true one to an acceptable accuracy as demonstrated in Figure 5. The angle error $\lambda$ between the two vectors is very small. Hence, it can be concluded that the error in $\alpha$ is small in the estimation process described above, and hence the error $\delta\mathbf{B}$ is small. This completes the process of computing the vector of the geomagnetic field $\mathbf{B}_{sdo}$, and completes the calculations needed in the proposed control algorithm.

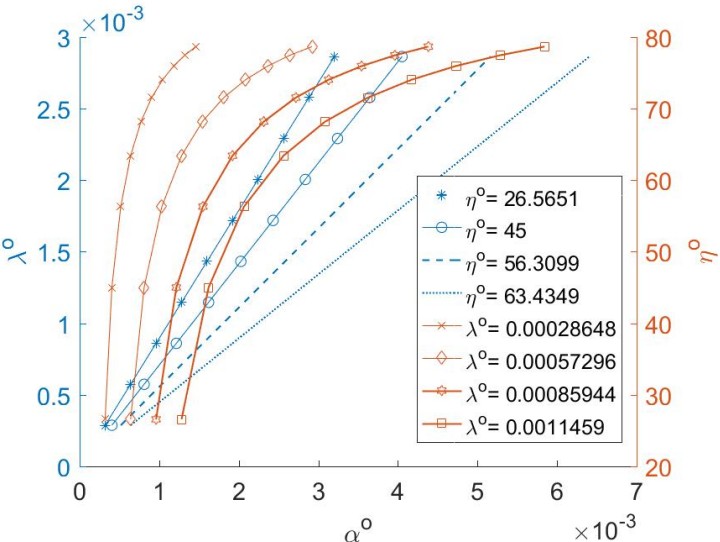

**Figure 4.** The relation between $\alpha$ and each of $\lambda$ (the left vertical axis) and $\eta$ (the right vertical axis). These results are obtained by changing the angles $\lambda$ and $\eta$ and the corresponding change in $\alpha$ is computed at a single sampling time of one of the simulation results.

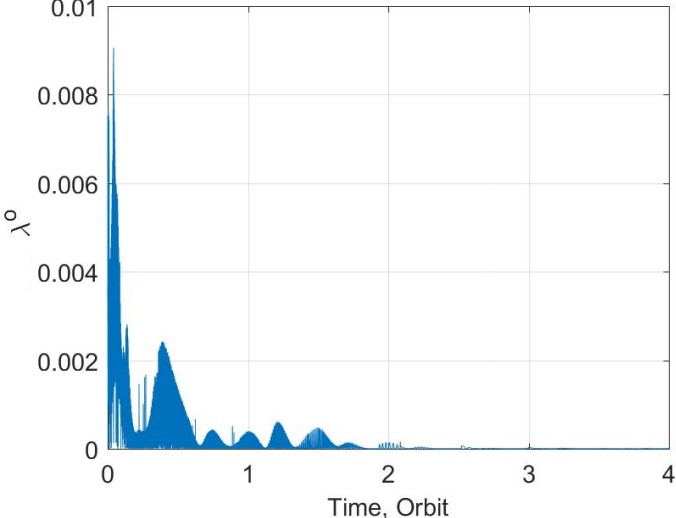

**Figure 5.** The angle $\lambda$, in degrees, between the actual torque **T** and the estimated torque $\tilde{\mathbf{T}}$ using the five-point stencil method for computing $\dot{\omega}$ numerically.

### 3.3. Attitude Estimation

The continuous-discrete MEKF in [59] is modified here to estimate the following: the attitude quaternion $\tilde{\mathbf{q}}$, the angular velocity $\tilde{\omega}$, and the magnetic field $\tilde{\mathbf{B}}$.

The measurements used by this MEKF are the angular velocity from the gyroscope, the direction of the sun from the sun sensor, and magnetic field pseudo measurements $\mathbf{B}_{sdo}$ at times $t_2$ and $t_3$, see Figure 2. For inertial pointing maneuvers, the state vector is $\mathbf{x}_k = [\omega_{bi}^T \; \mathbf{q}_{bi}^T \; \mathbf{B}^T]^T \in \mathbb{R}^{10}$, and the error state vector is $\delta\mathbf{x}_k = [\delta\omega_{bi}^T \; \delta\mathbf{q}_{v(bi)}^T \; \delta\mathbf{B}^T]^T \in \mathbb{R}^9$, where $\delta\mathbf{q}_v \in \mathbb{R}^3$ is the quaternion vector part error. Let $\mathbf{x}_{k|k-1}$ be the a prior estimate from the propagation step, and $\mathbf{x}_{k|k}$ be the posterior estimate from the update step. The propagation and update steps of the MEKF are presented below.

### 3.3.1. State Propagation

During the propagation step, the angular velocities and the quaternion state variables are integrated numerically, using a 4th order Runge–Kutta integration for the nonlinear kinetic Equation (3), and for the kinematic model in Equation (1), to obtain a prior estimate of the angular velocities $\boldsymbol{\omega}_{k|k-1}$ and the quaternion $\mathbf{q}_{k|k-1}$. The magnetic field state, however, will be propagated using a simple model as shown below.

It is assumed here, that the derivative of the **B** vector in the inertial frame does not change (i.e., $^i\dot{\mathbf{B}} = 0$) during the propagation of the magnetic field vector **B**. This assumption is acceptable for a small time step [9]. Consider the posterior estimated magnetic field vector $\mathbf{B}_{k-1|k-1}$, computed at the time step $k-1$. The quaternion conjugate of the posterior quaternion update at the time step $k-1$ is $\mathbf{q}_{k-1|k-1}^{-1}$. Then we can write:

$$\mathbf{B}_{k|k-1} \approx [R(\mathbf{q}_{k|k-1})]_{bi}[R(\mathbf{q}_{k-1|k-1}^{-1})]_{bi}\mathbf{B}_{k-1|k-1}, \tag{18}$$

where $[R(\mathbf{q}_{k-1|k-1}^{-1})]_{bo}$ is the transformation matrix from the body frame to the inertial frame at the time step $k-1$. The $^i\mathbf{B}_{k-1|k-1}$ can be transformed to the body frame using the transformation matrix $[R(\mathbf{q}_{k|k-1})]_{bi}$; hence for small time step we can assume that $^i\dot{\mathbf{B}}_k = 0$, and hence $^i\mathbf{B}_{k-1|k-1} \approx {}^i\mathbf{B}_{k|k-1}$; we can then write:

$$\begin{aligned}
\mathbf{B}_{k|k-1} &\approx [R(\mathbf{q}_{k|k-1})]_{bi}[R(\mathbf{q}_{k-1|k-1}^{-1})]_{bi}\mathbf{B}_{k-1|k-1} \\
&\approx [R(\mathbf{q}_{k|k-1})]_{bi}{}^i\mathbf{B}_{k-1|k-1} \\
&\approx [R(\mathbf{q}_{k|k-1})]_{bi}{}^i\mathbf{B}_{k|k-1}
\end{aligned} \tag{19}$$

Equation (19) can be used to propagate the magnetic field vector to acquire a prior estimate of the magnetic field $\mathbf{B}_{k|k-1}$ at step time $k$. The covariance matrix $[P]$ propagates in time according to the Joseph form [59], which has been shown to be numerically stable but requires more computational power [59]. In the interest of having a simpler numerical implementation that requires less computational power, the following approximation is adopted [59,60]:

$$[P]_{k|k-1} = [\phi_{t_k,t_{k-1}}][P_{k-1|k-1}][\phi_{t_k,t_{k-1}}]^T + [Q_{k-1}], \tag{20}$$

where $[Q_{k-1}]$ is the discrete-time process covariance matrix, and $[\phi_{t_k,t_{k-1}}]$ is the state transition matrix. In order to find the state transition matrix $[\phi_{t_k,t_{k-1}}]$, a linearization for this model is carried out as detailed below.

The derivative of Equation (19) is as follows:

$$\begin{aligned}
\dot{\mathbf{B}} &\approx [R(\mathbf{q})]_{bi}{}^i\dot{\mathbf{B}} + [\dot{R}(\mathbf{q})]_{bi}{}^i\mathbf{B} \\
&\approx -[\boldsymbol{\omega}_{bi}]_x[R(\mathbf{q})]_{bi}{}^i\mathbf{B} \\
&\approx \mathbf{B} \times \boldsymbol{\omega}_{bi},
\end{aligned} \tag{21}$$

where $[\dot{R}(\mathbf{q})]_{bi} = -[\boldsymbol{\omega}_{bi}]_x[R(\mathbf{q})]_{bi}$ [49]. It is worth noting that the same derivative equation can be obtained using the transport theorem as follows (assuming $^i\dot{\mathbf{B}} = 0$) [49,61,62]:

$$\begin{aligned}
\dot{\mathbf{B}} &= [R(\mathbf{q})]_{bi}{}^i\dot{\mathbf{B}} + [\mathbf{B}]_x \boldsymbol{\omega}_{bi} \\
&\approx \mathbf{B} \times \boldsymbol{\omega}_{bi}
\end{aligned} \tag{22}$$

The linearized first-order Taylor expansion version of Equation (22) results in the small-signal dynamic equation of the magnetic field as follows:

$$\delta\dot{\mathbf{B}}(t) \approx [\bar{\mathbf{B}}]_x \,\delta\boldsymbol{\omega}_{bi} - [\bar{\boldsymbol{\omega}}_{bi}]_x \,\delta\mathbf{B} \tag{23}$$

Using Equation (3), the small-signal dynamic equation of the angular velocity becomes:

$$
\begin{aligned}
\delta\dot{\boldsymbol{\omega}}_{bi}(t) &\approx \left[[I]^{-1}[[I\bar{\boldsymbol{\omega}}_{bi}]_x - [\bar{\boldsymbol{\omega}}_{bi}]_x[I]]\right]\delta\boldsymbol{\omega}_{bi} \\
&+ [I]^{-1}([\bar{\mathbf{M}}]_x\delta\mathbf{B} - [\bar{\mathbf{B}}]_x\delta\mathbf{M})
\end{aligned}
\tag{24}
$$

The quaternion small-signal dynamic equation is:

$$
\delta\dot{\mathbf{q}}_{v(bi)}(t) \approx 0.5[\mathbf{1}_{3x3}]\delta\boldsymbol{\omega}_{bi} - [\bar{\boldsymbol{\omega}}_{bi}]_x\delta\mathbf{q}_{v(bi)}
\tag{25}
$$

The state transition matrix can be approximated as $[\phi] \approx [\mathbf{1}_{9x9}] + [F(x)]\,dt$ for small time step $dt$ [63], where $[F(x)]$ is the Jacobian matrix. The Jacobian matrix can be computed from Equations (23)–(25) to give:

$$
[F(x)] = \begin{bmatrix} F_{11} & [\mathbf{0}_{3x3}] & [I]^{-1}[\bar{\mathbf{M}}]_x \\ 0.5[\mathbf{1}_{3x3}] & -[\bar{\boldsymbol{\omega}}_{bi}]_x & [\mathbf{0}_{3x3}] \\ [\bar{\mathbf{B}}]_x & [\mathbf{0}_{3x3}] & -[\bar{\boldsymbol{\omega}}_{bi}]_x \end{bmatrix},
\tag{26}
$$

where

$$
F_{11} = [I]^{-1}[[I\bar{\boldsymbol{\omega}}_{bi}]_x - [\bar{\boldsymbol{\omega}}_{bi}]_x[I]],
\tag{27}
$$

where the nominal values $[\bar{\cdot}]$ are the a priori propagated values of the state vector.

### 3.3.2. State Update

A linearization of the measurement model about the a priori state estimate is here carried out for use inside the MEKF. The sun sensor measurement is $\mathbf{V}_{sun}$, where:

$$
\mathbf{V}_{sun(k|k-1)} = [R(\mathbf{q}_{k|k-1})]\,^{i}\mathbf{V}_{sun(k)},
\tag{28}
$$

Assuming small angles, the transformation matrix can be approximated as $[R(\mathbf{q})] = [R(\bar{\mathbf{q}})][R(\delta\mathbf{q})] \approx [R(\mathbf{q}_{k|k-1})]([\mathbf{1}_{3x3}] - 2[\delta\mathbf{q}_v]_x)$. Using this approximation, the error in the sun direction can be approximated as follows:

$$
\mathbf{V}_{sun(k|k-1)} - \bar{\mathbf{V}}_{sun(k|k-1)} \approx 2[\bar{\mathbf{V}}_{sun(k|k-1)}]_x\,\delta\mathbf{q}_v
\tag{29}
$$

The linearized small error measurement model about the apriori state estimate can be written as follows:

$$
\mathbf{Z}_k = [H_k]\,\delta\mathbf{x} = \begin{bmatrix} [\mathbf{1}_{3x3}] & [\mathbf{0}_{3x3}] & [\mathbf{0}_{3x3}] \\ [\mathbf{0}_{3x3}] & 2[\bar{\mathbf{V}}_{sun(k|k-1)}]_x & [\mathbf{0}_{3x3}] \\ [\mathbf{0}_{3x3}] & [\mathbf{0}_{3x3}] & [\mathbf{1}_{3x3}] \end{bmatrix}\delta\mathbf{x}
\tag{30}
$$

where $\mathbf{Z}_k$ is the small-signal (error) measurement vector. At each measurement time, a Kalman gain is computed using Equation (31).

$$
[K_k] = [P_{k|k-1}][H_k]^T\left([H_k][P_{k|k-1}][H_k]^T + [R_k]\right)^{-1}
\tag{31}
$$

The states error vector $\delta\mathbf{x}$ is computed as follows:

$$
\delta\mathbf{x} = \begin{bmatrix} \delta\boldsymbol{\omega}_{k|k} \\ \delta\mathbf{q}_v \\ \delta\mathbf{B}_{k|k} \end{bmatrix} = [K_k]\begin{bmatrix} \boldsymbol{\omega}_{mes} - \boldsymbol{\beta} - \boldsymbol{\omega}_{k|k-1} \\ \hat{\mathbf{V}}_{sun(mes)} - \hat{\mathbf{V}}_{sun(k|k-1)} \\ \mathbf{B}_{sdo} - \mathbf{B}_{k|k-1} \end{bmatrix},
\tag{32}
$$

where $\boldsymbol{\omega}_{mes}$ and $\hat{\mathbf{V}}_{sun(mes)}$ are the measurements from the gyroscope and sun sensor respectively. The $\boldsymbol{\beta}$ is the bias vector that will be estimated using the equations of the $\text{EKF}_\omega$ that are given in Appendix A. The update step is carried out for each of the $\mathbf{q}$, $\boldsymbol{\omega}$,

and **B** states differently. The quaternion is updated using a quaternion multiplication as shown in Equation (33).

$$\mathbf{q}_{k|k} = \begin{bmatrix} \sqrt{1 - \|\delta\mathbf{q}_v\|^2} \\ \delta\mathbf{q}_v \end{bmatrix} \otimes \mathbf{q}_{k|k-1}, \tag{33}$$

where $\otimes$ represents the quaternion product. The term $\sqrt{1 - \|\delta\mathbf{q}_v\|^2}$ is used for preserving the quaternion normalization of the computed quaternion error $\delta\mathbf{q}$, as shown in Equation (33).

The magnetic field **B**, on the other hand, is updated in two steps. Recall that the propagation step of **B** in Equation (19) used $\mathbf{q}_{k|k-1}$. Now that $\mathbf{q}_{k|k}$ is available, the latter is used to achieve a better propagation of **B**. This is carried out as follows: $\mathbf{B}^+_{k|k-1} = [R(\delta\mathbf{q}_k)]\mathbf{B}_{k|k-1}$. Then the magnetic field is updated using this new propagated vector $\mathbf{B}^+_{k|k-1}$ along with the error in magnetic field vector $\delta\mathbf{B}_{k|k}$, which is computed using the associated part of the Kalman gain and the measurements (or pseudo measurements), Equation (32):

$$\mathbf{B}_{k|k} = \mathbf{B}^+_{k|k-1} + \delta\mathbf{B}_{k|k} \tag{34}$$

$$\boldsymbol{\omega}_{k|k} = \boldsymbol{\omega}_{k|k-1} + \delta\boldsymbol{\omega}_{k|k} \tag{35}$$

where the spacecraft angular velocity is updated using the standard approach in Equation (35).

The estimated angular velocity bias vector (in the intervals $t_i - t_{i+1}$) from $\text{EKF}_\omega$ is used here to update the angular velocity and will not be estimated at times $t_i$ to reduce the computational cost. The estimation error covariance matrix $[P]$ is updated as follows:

$$[P_{k|k}] = ([\mathbf{1}_{9x9}] - [K_k][H_k])[P_{k|k-1}] \tag{36}$$

*3.4. Stability and Performance of the Modified MEKF*

The estimation error covariance matrix $[P]$, which could turn into a non-positive matrix throughout the propagation and update processes, is one of the elements that affects the stability of the MEKF. Numerical instabilities could cause this. In order to confirm the stability, the eigenvalues of the matrix $[P]$ are always confirmed to be positive during the simulation of the Monte Carlo runs in the following section.

Performance of the filter is significantly influenced by the kind of measurement and process noise errors. The performance of the filter might be substantially hampered by inaccurate noise error representation. In order to obtain the best approximation, a filter tuning method is typically used to modify the filter's parameters. It is possible to tune filters offline using numerical optimization methods or online using adaptive algorithms [64]. However, according to [64], manual optimization is more common in practise. To give the measurements the maximum weight possible during manual optimization, choose small values for the measurement error covariance matrix, smaller than what the manufacturer or a laboratory test supplied. After that, modify the original estimate and process the noisy covariance matrices to obtain an acceptable level of performance. The method is then repeated until the needed performance is achieved, as noted in [64]. According to the statistical analysis of the outcomes of the Monte Carlo simulation, a manual optimization approach for the covariance matrices is used and validated in this study.

**4. Numerical Simulations**

The goal of this section is to compare the proposed algorithm in this paper, which is shown in Figure 2, to a reference standard algorithm. In the reference algorithm, both the magnetic rods and the magnetometers are turned on, alternately, during each cycle period, as shown in Figure 1. Monte Carlo (MC) simulation is conducted for this comparison. To highlight the impact of the proposed algorithm compared to the reference one, the output results from the proposed algorithm are normalized by the results from the reference algorithm. The hardware configuration and spacecraft parameters are the same for both

algorithms and are similar to those in reference [2]. Table 1 shows both the spacecraft parameters and sun-synchronous orbital parameters.

**Table 1.** Spacecraft and orbital parameters.

| Parameter | Value [Unit], [Uncertainty] |
|---|---|
| $[I_x, I_y, I_z]^T$ | $[0.196, 0.202, 0.202]^T$ [kg $\cdot$ m$^2$] , [10%] |
| Max. dipole moment vector | $\pm[1.83 \ 1.83 \ 1.83]^T$ [Am$^2$] |
| Altitude | 639.212 [km] |
| Inclination | 97.868° |
| Right ascension of the ascending | 157.305° |
| True anomaly at initial time | 277.29° |
| Eccentricity | 0 |

The orbital position and velocity of the spacecraft are propagated in time using a model that accounts for the J2 gravitational effect. For orbit propagation, the J2000 inertial frame of reference is used. The orbit propagator output is used to calculate the spacecraft's location with regard to the earth and sun, as well as the aerodynamic density and geomagnetic field.

To replicate the uncertainty in the spacecraft and environment models, a random Gaussian process is applied. An additional Gaussian random-direction torque is provided to address unknown torque sources, modeling flaws in the gravity gradient torque model, and inertia uncertainty in Equation (3). The mean value of this torque is chosen to be zero, with a standard deviation of $1 \times 10^{-9}$ [N $\cdot$ m]. The uncertainty in the spacecraft inertia tensor when computing the torque in Section 3.2 is modeled as an additive Gaussian variable with a mean of 10%. The simulation parameters are presented in Table 2. The measurements of the sun sensor and the magnetometers are assumed to have random Gaussian noises and a static bias. The mean value of this white noise is adjusted to zero for both the sun sensor and the magnetometer, with standard deviations of $1 \times 10^{-4}$ and $5 \times 10^{-3}$ [Tesla], respectively. The gyroscope is assumed subject to a drift bias and stochastic white noise. This white noise's mean value is set to zero, with a standard deviation of $1 \times 10^{-4}$ [rad/s].

**Table 2.** Disturbance Parameters.

| Parameter | Value [Unit] | Uncertainty |
|---|---|---|
| $\rho$ * | $2.01 \times 10^{-14}$ [kg $\cdot$ m$^3$] | – |
| $R_{mp}$ | [9 11 12] [mm] | 10% |
| $C_D$ | 2 | – |
| Spacecraft dimension | [23 23 29] [cm] | – |
| $\|\mathbf{M}_{rds}\|$ | $1 \times 10^{-4}$ [Am$^2$] | 10% |
| $C_{rk}$ | 1.5 | – |
| $F_{solar}$ | 1366 [w/m$^2$] | – |

* $\rho$ is computed using an empirical formula provided in http://www.braeunig.us/space/atmmodel.htm (accessed date 17 November 2022).

The parameters of the control algorithms are as follows: $T_s = 0.25$ [s] and $\delta = 0.7$. Therefore, the control command frequency is $f_c = 1/0.25$, and the gyroscope measurements frequency is selected to be $f_\omega = 5/0.25$. The control gains are: $\zeta = 0.001$, $k_p = 0.01$, and $k_d = 0.4$. A confirmation window of 5-min is utilized after settling to the specified attitude.

*4.1. Case Study*

Before presenting the statistical MC analysis, the results from a sample run for the proposed ADCS algorithm are presented. In this example, each of the satellite's initial angular velocity (true and estimated), and the initial attitude (true and estimated) are selected randomly. The simulation runs for 10 orbits. To make this presentation more clear, the attitude error will be represented in terms of the principal rotation error angle $\phi$ between the current attitude and the desired one. Figure 6 shows the principal rotation

error angle for the reference case and the proposed algorithm with $\epsilon = 6$ , (labeled "Ref" and "Proposed $\epsilon = 6$" respectively). As shown in Figure 6, the proposed algorithm is able to settle at the desired attitude faster than the reference algorithm. Further notice, as can be shown for the zoom part of Figure 6, that the average attitude steady-state error of the proposed algorithm is less than the one from the reference case. This is due to increasing the duty cycle of magnetic rods that leads to decreasing the steady state error as analyzed in [6].

Figure 7 depicts the time behavior of the satellite's angular velocity magnitude of the body frame. The figure shows that almost a zero angular velocity magnitude is preserved by the proposed algorithm after settling the spacecraft at the specified attitude. Figure 8 represents the magnetic field time history of the estimated and true values. As are often seen, a good match has been achieved. Later, six parameters will be presented to assess the ambient magnetic field estimation performance.

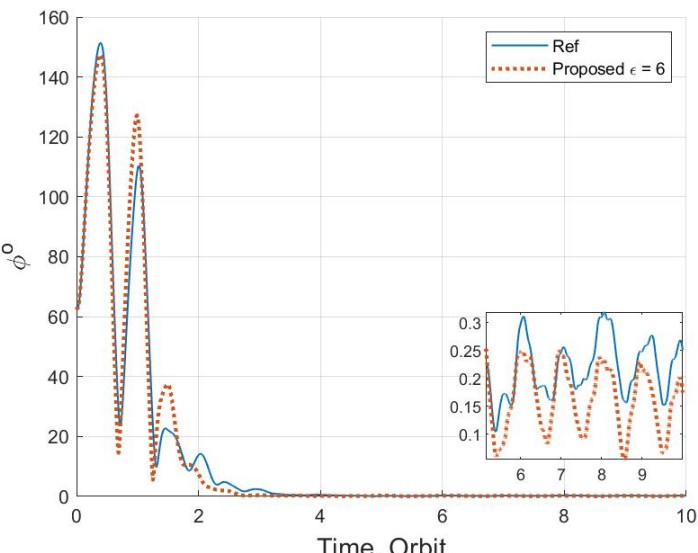

**Figure 6.** Attitude error history in terms of principal rotation angle $\phi$ between the real and desired attitude.

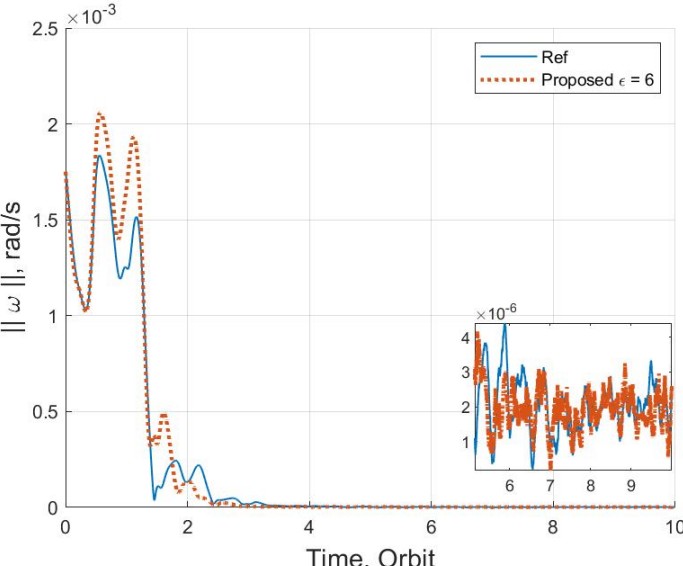

**Figure 7.** Spacecraft angular velocity magnitude history.

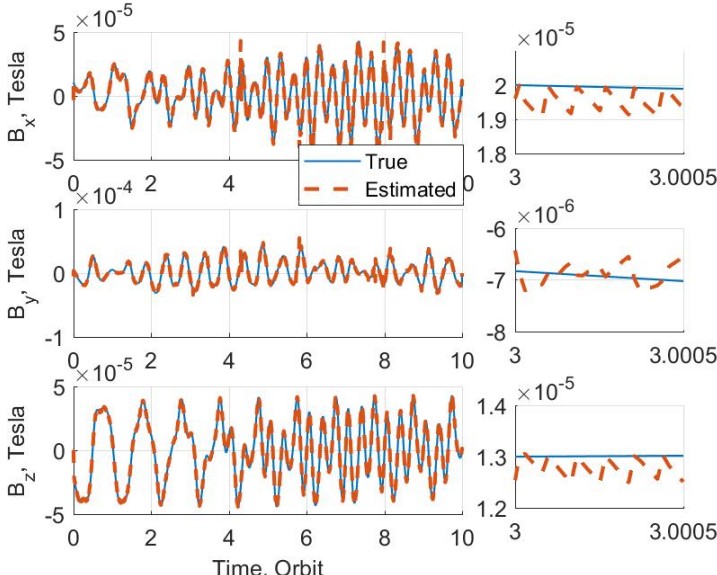

**Figure 8.** Ambient magnetic field history over time.

The above simulation is repeated for several $\epsilon$ values of the proposed algorithm. In this section, two parameters are used for comparison. The first parameter is the Power Consumption (PC) of the magnetic rods. The second parameter is the Maneuver Time (MT), which is defined here as the time until the spacecraft's attitude is less than $1°$ error around the desired attitude plus the confirmation window of five minutes in which the spacecraft attitude remains within the error bounds. Regarding the Computational Load (CL) of the proposed algorithm, a measure of the total computational time per maneuver is computed using the Matlab built-in function (tic/toc). The above three parameters are computed for the proposed algorithm, and then are normalized w.r.t. the same parameters of the reference algorithm.

Figure 9 shows the normalized maneuver time ($N(MT)$), the normalized power consumption ($N(PC)$) (on the left axis) and the normalized computational load ($N(CL)$) (on the right axis,) for different values of $\epsilon$. The $N(PC)$ and $N(MT)$ are always less than 1, which means less power consumption and less maneuver time compared to the reference algorithm, for all $\epsilon$. Both the PC and the MT improve (decrease) as $\epsilon$ increases, up to a point. As $\epsilon$ increases beyond the value of 6, the change in $\delta_{eqv}$ becomes very small. For example, the proposed duty cycle is $\delta_{eqv} = 0.95$ at $\epsilon = 6$, while at $\epsilon = 10$, it is $\delta_{eqv} = 0.97$, see Equation (5). Increasing $\epsilon$, however, increases the CL as shown on the right vertical axis in Figure 9.

For further assessment of estimating the ambient magnetic field, six parameters are used. The first is the correlation coefficient (CC) between the estimated and true values of the magnetic field. When the CC is close to 1, it indicates a strong correlation, and when the CC is close to 0, it indicates weak correlation. The scatter index (SI) indicates statistically how the computed quantity is scattered around the true one, the smaller the SI the higher is that the performance. The normalized root mean square error (NRMSE), mean absolute error (MAE), bias, and root mean square error (RMSE) are also computed for the obtained simulation data. Figure 10 shows the change of the CC with $\epsilon$ on the left axis. On the right axis, the NRMSE and SI are depicted. The CC is close to 1, and the SI and NRMSE are very small, indicating a strong correlation with less scattering between the estimated and the true magnetic fields. Figure 11 shows the bias, RMSE, and MAE. The variation of all parameters confirms that the lower the $\epsilon$ the better ambient magnetic field estimation. In the following Monte Carlo analysis, the $\epsilon$ value is selected to be 6 which means $\delta_{eqv} = 0.95$ compared to $\delta = 0.7$ in the reference algorithm.

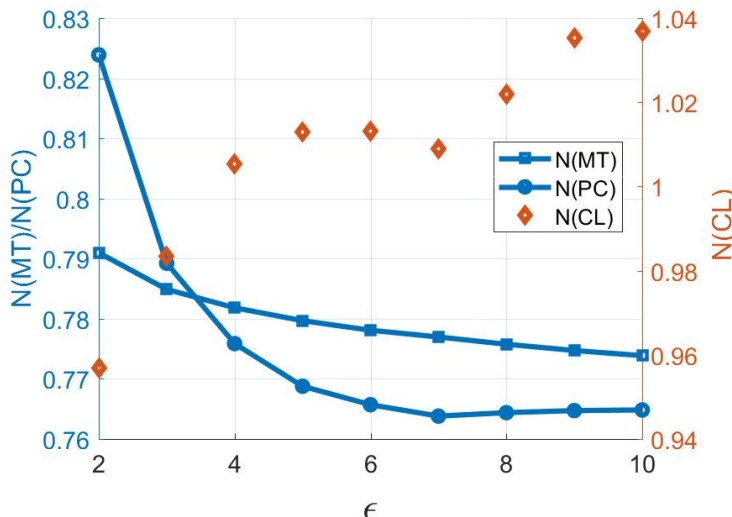

**Figure 9.** N(MT), N(PC), and N(CL) versus $\epsilon$.

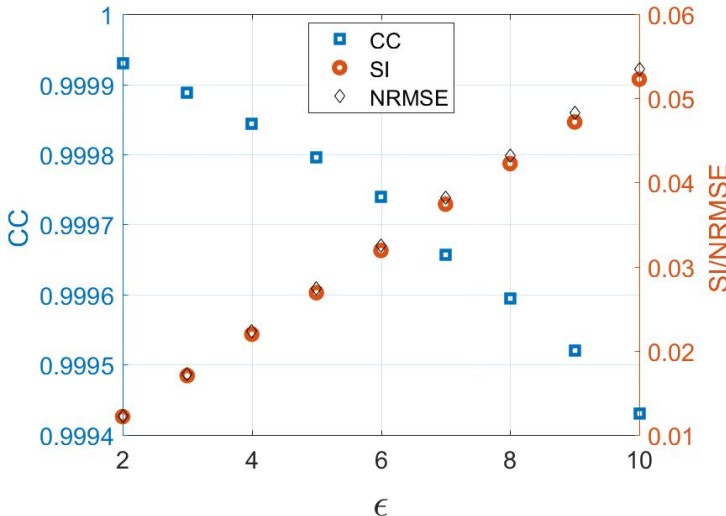

**Figure 10.** CC, SI and NRMSE of the estimated $\tilde{\mathbf{B}}$ vector.

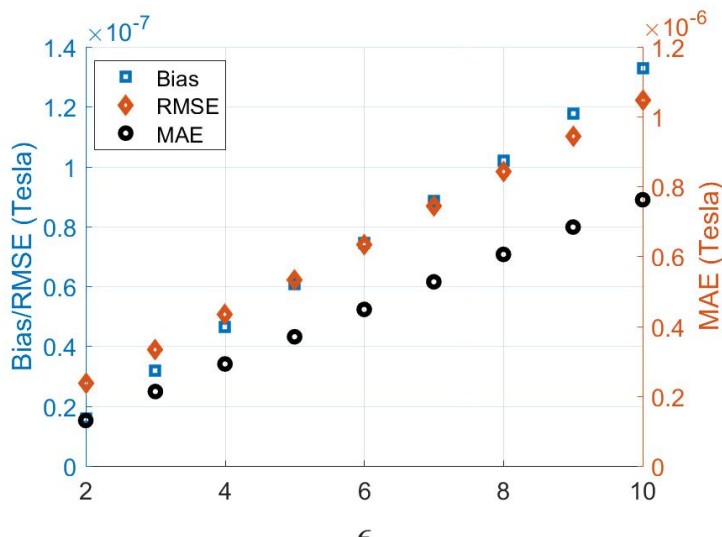

**Figure 11.** Bias, MAE and RMSE of the estimated $\tilde{\mathbf{B}}$ vector.

### 4.2. Monte Carlo Simulation Analysis

Results are here presented for 45,000 Monte Carlo runs; 22,500 for the reference algorithm and 22,500 for the proposed algorithm. Normal distribution is used for noises seeds, with a different distribution for each run.

The 22,500 categories have different initial angular rates (true and estimated), and initial quaternion (true and estimated), from all other categories. These values are generated randomly and are equivalent for both algorithms. So, the results are reported in terms of improvement percentage in PC, MT, and CL. Here, the percent improvement $P_{prm}$ is computed as follows:

$$P_{prm} = 100 \times \left( 1 - \frac{prm_P}{prm_R} \right), \tag{37}$$

where *prm* represents the MT or the PC or the CL, the subscript *P* represents the computed *prm* values for the proposed ADCS, the subscript *R* represents the computed *prm* values for the reference ADCS, and $P_{prm}$ represents the percentage improvement in the *prm*. As a measure of the estimation error, the CC, SI, NRMSE, MAE, Bias, and RMSE are computed and averaged. All the disturbance torques mentioned in this paper are simulated. Table 3 shows the estimate of the magnitude of these disturbance torques, in the worst-case of the entire MC runs.

**Table 3.** Worst-case disturbance torque magnitudes.

| Disturbance | Magnitude [N · m] |
|---|:---:|
| Aerodynamic drag | $5.19 \times 10^{-9}$ |
| Gravity gradient | $1.04 \times 10^{-8}$ |
| Residual dipole | $4.06 \times 10^{-9}$ |
| Solar radiation | $1.07 \times 10^{-9}$ |

Figure 12 depicts the Gaussian distribution and histogram of $P_{MT}$. Figure 12 shows that the maneuver time of the proposed algorithm is significantly less than that of the reference case. In some cases, the proposed algorithm achieves $P_{MT}$ of about 50%, whereas in other cases there is almost no improvement, compared to the reference algorithm. The mean value of the $P_{MT}$ is 23.17%, with 8.53% standard deviation.

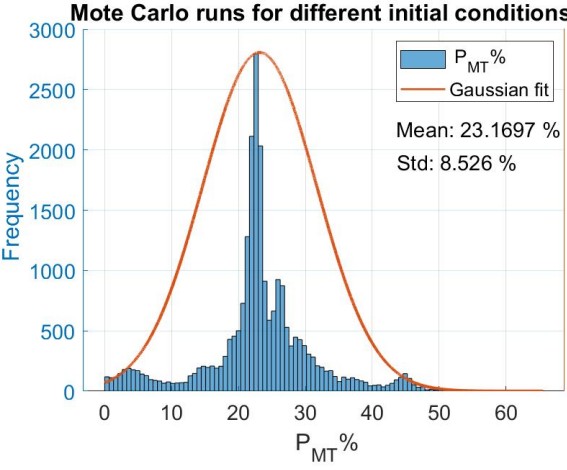

**Figure 12.** Gaussian distribution and histogram of improvement percentage in maneuver time.

In terms of power consumption, the proposed algorithm increases the mean value of the $P_{PC}$ significantly as shown in Figure 13. The $P_{PC}$, using the proposed algorithm, goes up to about 45% in some cases. The mean value for the $P_{PC}$ is 19.62%, with a 10.63% standard deviation.

The computational load is computed for the whole maneuver. Overall, the proposed algorithm has a higher computational load compared to the reference algorithm because of the additional computations in evaluating $\mathbf{B}_{sdo}$ and in the MEKF magnetic propagation step. It is noticed that the computational load of the proposed algorithm roughly needs about 30% more computational resources at every time step. It should be noted that, while this approach provides an approximate estimate of the CL when run on the Matlab environment, it does not explain the CL of the suggested technique when run on flying hardware. In reality, the technique may be considerably more effective in terms of CL after implementation than Matlab suggested, since the CL may be substantially improved throughout the implementation.

However, when the savings in the maneuver time are significant, the computational time of the proposed algorithm becomes less than that of the reference algorithm, simply because the whole maneuver is completed in a significantly shorter period of time, and hence the computations stop much sooner compared to the reference algorithm. This observation is evident in Figure 14, where there is a strong correlation between the $P_{MT}$ and the $P_{CL}$.

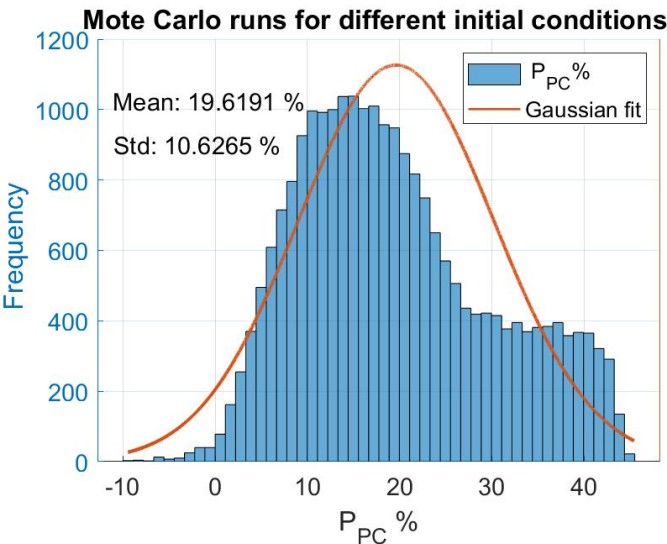

**Figure 13.** Gaussian distribution and histogram of improvement percentage in power consumption.

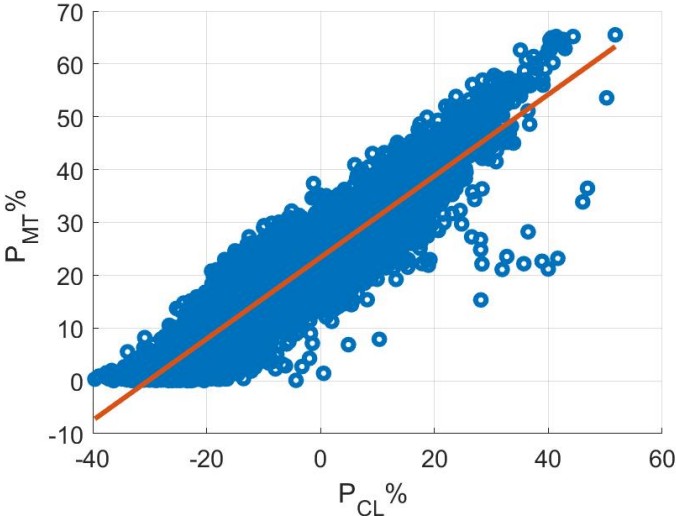

**Figure 14.** Improvement percentage in computational load versus Improvement percentage in maneuver time. The tendency of the CL is shown by the red average line.

Finally, regarding the estimated magnetic field parameters within the above MC simulations, all six parameters are computed as follows. Each one run resulted in a time history for $\tilde{\mathbf{B}}$ and $\mathbf{B}$ vectors. The six parameters are computed for each component of $\tilde{\mathbf{B}}$ resulting in three components. For each run, an average of the three components is computed.

The mean and standard deviation values for the six parameters for the entire MC runs are presented in Table 4. The results show a strong correlation and less scattering in these MC runs.

**Table 4.** Mean and standard deviation of the parameters used for comparison for the entire MC runs.

| Parameters | Mean | Standard Deviation |
|---|---|---|
| $P_{MT}$ | 19.62% | 10.63% |
| $P_{PC}$ | 23.17% | 8.53% |
| $P_{CL}$ | −0.24% | 10.13% |
| MAE | $4.36 \times 10^{-7}$ [Tesla] | $9.366 \times 10^{-9}$ [Tesla] |
| Bias | $7.861 \times 10^{-8}$ [Tesla] | $2.063 \times 10^{-8}$ [Tesla] |
| RMSE | $6.428 \times 10^{-7}$ [Tesla] | $4.084 \times 10^{-8}$ [Tesla] |
| NRMSE | 0.0343 | 0.0022 |
| SI | 0.0339 | 0.002 |
| CC | 0.99 | 0.0002 |

## 5. Verification Using Real Data

This section is devoted to validating the estimation of the magnetic field. Real telemetry data from the CASSIOPE spacecraft are used for verification. The CASSIOPE is a multimission satellite from Canadian Space Agency (CSA). Its objectives are space weather operation and high-speed communications concepts verification [44,65]. Three-star trackers are used for attitude determination. Two magnetometers are installed on two different booms. The magnetic rods' maximum dipole moment is 30 [Am$^2$] and they have activated alternately with the magnetometers, with a duty cycle of $\delta = 0.7$.

The technical team provided real telemetry information for the ground station tracking maneuvers, that is performed on 21 February 2019. These telemetry data consists of the time history of angular velocity measurements $\boldsymbol{\omega}$, magnetic field measurements $\mathbf{B}$, reaction wheels torques $\mathbf{T}_w$, magnetic rods torques $\mathbf{T}$, controller dipole moment $\mathbf{M}$, control term or designed torque $\mathbf{T}_{req}$ and Ephemeris. The designed torque $\mathbf{T}_{req}$ is the magnetic controller output. The Ephemeris includes the attitude (in terms of the Euler angles roll, pitch and yaw,) the spacecraft position and velocity in an inertial frame, latitude, longitude, and altitude (in the World Geodetic System 1984 (WGS84) frame, and in the Earth-centered inertial (J2000) frame). The telemetry data sampling time is 0.1 s, whereas the Ephemeris sampling time is 5 s, for two maneuvers with periods 200 s and 360 s, respectively. The duty cycle will be $\delta = 0.7 \times 0.1 = 0.07$ s as a result.

The star tracker provides the attitude. Therefore, no attitude estimation is performed in this verification process. However, this attitude information will be used in Equation (18) to propagate the magnetic field.

The CASSIOPE is modeled in this paper as a rigid body, and its moments of inertia are optimized to account for unmodeled structural flexibility. The initial inertia tensor matrix is given as follows:

$$[I] = \begin{bmatrix} 186.5202 & -0.6839 & -5.2728 \\ -0.6839 & 194.4095 & 4.2445 \\ -5.2728 & 4.2445 & 214.1428 \end{bmatrix} [\text{kg} \cdot \text{m}^2] \tag{38}$$

The magnetic field estimation verification process is as follows:

(1)  Computing the pseudo measurement $\mathbf{B}_{sdo}$ as described in Section 3.2. The spacecraft angular velocities provided by the gyroscope have bias and noise. Hence a batch

optimization process is first conducted to estimate this bias. This optimization process searches for the spacecraft's initial angular velocity that minimizes the difference between the true attitude $\mathbf{q}_t$ (obtained from the star trackers) and the propagated one $\mathbf{q}_{prog}$ (obtained using the attitude kinematics Equation (1)). The objective function $J$ is:

$$J = \int_{t_0}^{t_f} \|\mathbf{q}_{prog} - \mathbf{q}_t\| dt \tag{39}$$

The bias vector is here assumed to be the difference between the mean value of the propagated angular velocity (using the initial angular velocity output from the optimization process) and the measured angular velocity from the gyroscope (the mean value of a polynomial fit for the measurements). This step is performed on just one occasion only. Therefore the output unbiased angular velocity measurements are shown in Figures 15 and 16, (labeled "Unbiased Measurement").

(2) An Extended Kalman Filter (EKF) is used to estimate the noises from these unbiased angular velocity measurements. The EKF uses the following three diagonal covariance matrices: initial state covariance $[P] = diag(1\,1\,1) \times 10^{-5}$, measurement noise error covariance $[R] = diag(1\,1\,1) \times 10^{-2}$, and process noise error covariance $[Q] = diag(1\,1\,1) \times 10^{-5}$, where $diag(x)$ means a diagonal matrix whose diagonal elements are $x$. The covariance matrices are chosen manually, as indicated in the Section 3.4, which follows the procedure provided in [64]. Euler Equation (3) is used for propagating the spacecraft angular velocity in the EKF, while the unbiased angular velocity is used as the measurement input to the EKF. The output angular velocity from EKF is shown in Figures 15 and 16. The equations of this EKF are given in Appendix B.

(3) The numerical calculation of the spacecraft angular speed derivative $\dot{\omega}$ using the five-stencil approach requires at least five consecutive angular velocity readings, in one cycle period (in which the torque is constant). Therefore, interpolation is employed to compute these velocities. Once the $\dot{\omega}$ is computed numerically, the pseudo measurement $\mathbf{B}_{sdo}$ is computed as discussed in Section 3.2.

(4) Another EKF is used for magnetic field estimation, where the pseudo measurement $\mathbf{B}_{sdo}$ is the EKF input measurement. The magnetic field propagation model Equation (18) propagates the magnetic field. The following three diagonal covariance matrices are used: initial state covariance $[P] = diag(2\,2\,2) \times 10^{-6}$, measurement noise error covariance $[R] = diag(0.5\,0.5\,0.5) \times 10^{-4}$, and $[Q] = diag(0.5\,0.5\,0.5) \times 10^{-8}$ is used as the process noise error covariance. The equations of this EKF are given in Appendix C.

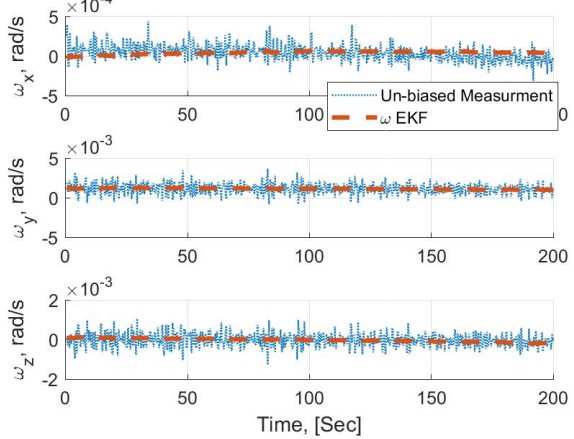

**Figure 15.** Angular velocities history from unbiased gyroscope measurements and the EKF output for 1st maneuver.

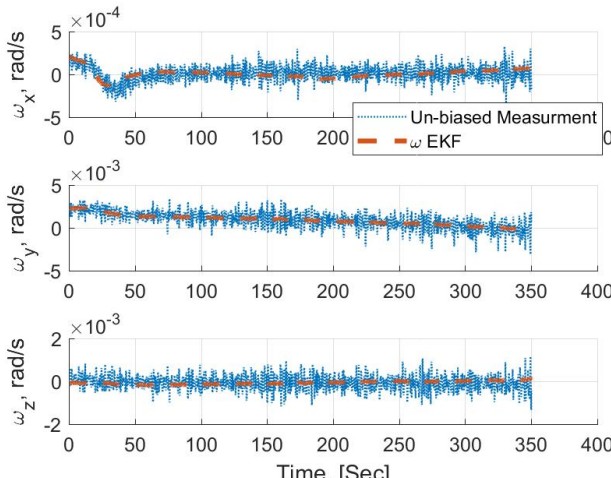

**Figure 16.** Angular velocities history from unbiased gyroscope measurements and the EKF output for 2nd maneuver.

It is worth noting that the two EKFs can be combined together. However, they are implemented separately in this study.

For the first 200 s maneuver, using $\epsilon = 5$, Figure 15 shows the un-biased angular velocity measurement versus the estimated one using the EKF. The matching proves that the rigid body Euler model can reasonably render the CASSIOPE dynamics. Figure 17 shows the comparison between the X components of each of the real magnetic field measurement (labeled "True"), the pseudo measurement $\mathbf{B}_{sdo}$ (labeled "Computed",) and the EKF estimated values (Labeled "Estimated").

Figures 17–19 render the good performance of the estimation process in the X, Y, and Z directions, respectively. The six-validation parameters are computed for different values of $\epsilon$ and the results are plotted in Figures 20 and 21. The results here are in agreement with the conclusions from the Monte Carlo analysis; the lower the $\epsilon$ the better the magnetic field estimation accuracy.

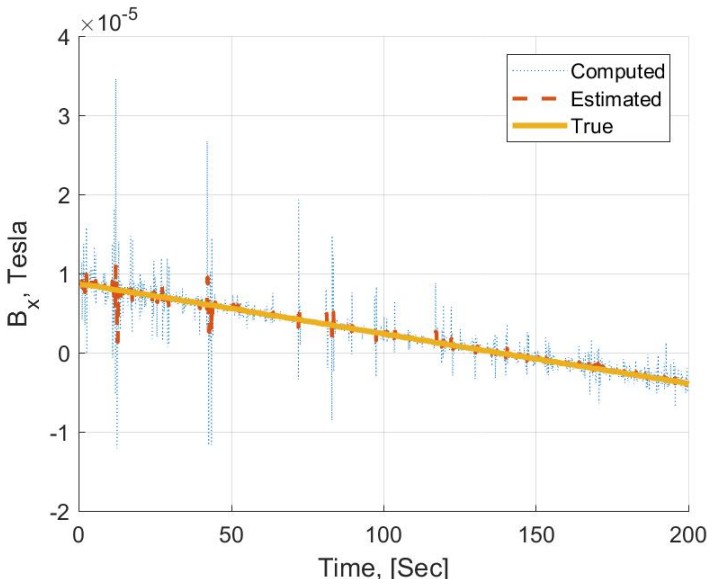

**Figure 17.** Magnetic field history in the X direction for 1st maneuver.

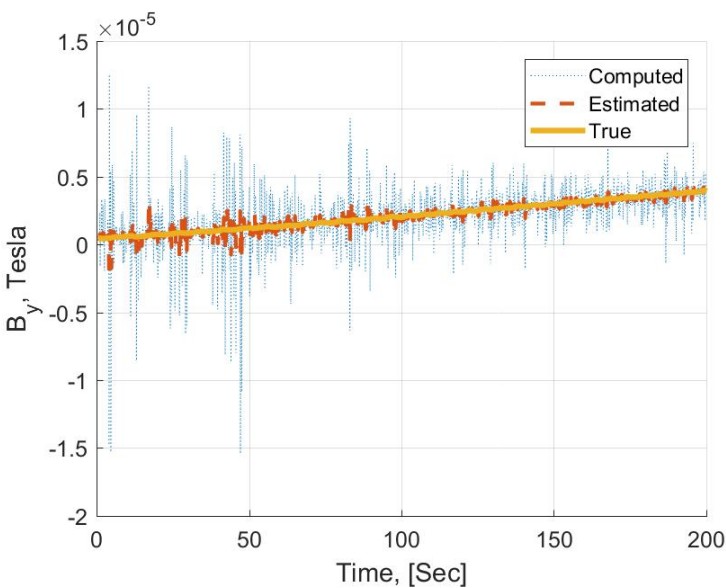

**Figure 18.** Magnetic field history in the Y direction for 1st maneuver.

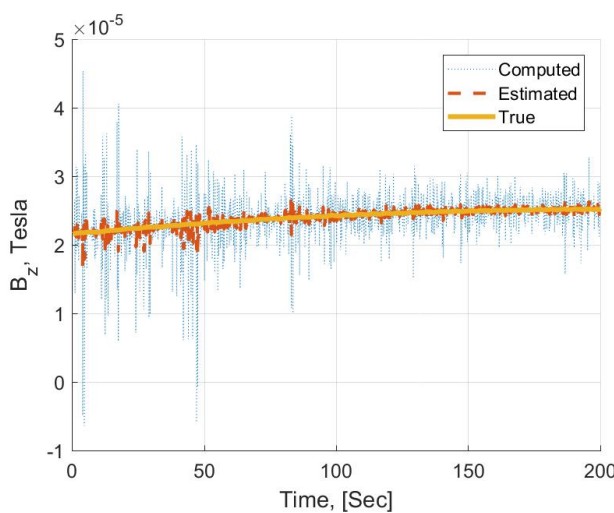

**Figure 19.** Magnetic field history in the Z direction for 1st maneuver.

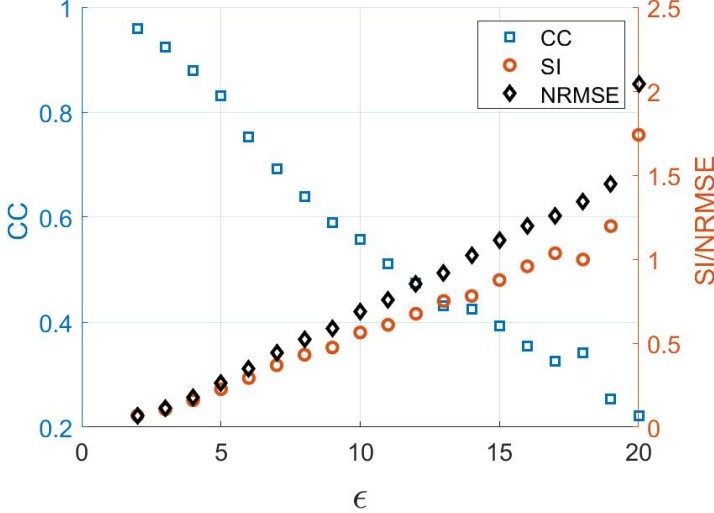

**Figure 20.** CC, SI and NRMSE of the estimated magnetic field for 1st maneuver.

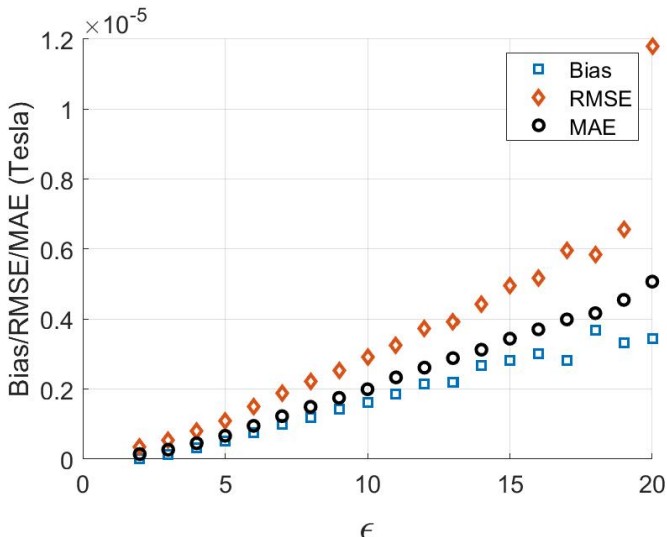

**Figure 21.** Bias, MAE and RMSE of the estimated magnetic field for 1st maneuver.

Another maneuver of 300 s duration is also verified. The angular velocities histories are shown in Figure 16. The magnetic field estimation values compared with the pseudo measurements and the true measurement are plotted in Figure 22. The magnetic field estimation accuracy is good, as evident from Figure 22. This is also confirmed by the values of the six validation parameters, which are listed in Table 5, for this maneuver, using $\epsilon = 5$.

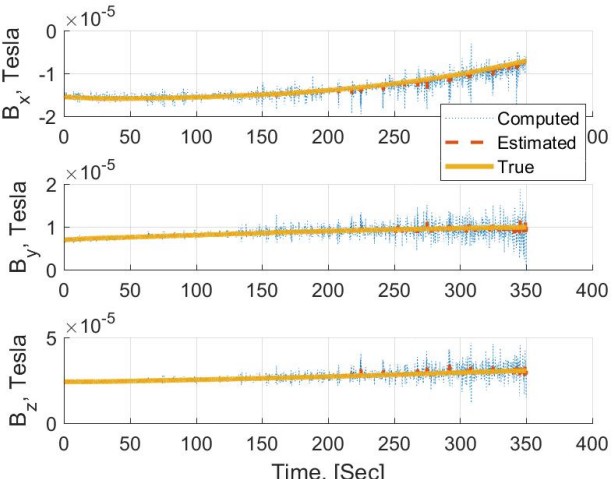

**Figure 22.** Magnetic field histories for the 2nd maneuver.

**Table 5.** Validation parameters for the second maneuvers.

| CC | SI | NRMSE | RMSE [Tesla] | MAE [Tesla] | Bias [Tesla] |
|---|---|---|---|---|---|
| 0.997 | 0.021 | 0.022 | $3.357 \times 10^{-7}$ | $4.852 \times 10^{-8}$ | $1.844 \times 10^{-7}$ |

## 6. Conclusions

An estimation algorithm was presented for spacecraft attitude that enables more efficient operation of the magnetic rods in inertial pointing attitude maneuvers. The proposed algorithm estimates the spacecraft attitude, in addition to the magnetic field, at the times when the magnetometer is not used. It was demonstrated via Monte Carlo simulations that the proposed algorithm results in shorter maneuver times of around 23% as well as less power consumption of around 19% by the magnetic rods. The results also show a less attitude steady state error due to increasing the duty cycle of the magnetic rods that are com-

patible with the analytical study in [6]. The magnetic field estimation process was tested against real data from the CASSIOPE mission and demonstrated good estimation accuracy.

**Author Contributions:** Methodology, M.A.A.D. and O.A.;software, M.A.A.D.; validation, M.A.A.D.; writing—original draft preparation, M.A.A.D.; writing—review and editing, O.A.; Supervision, O.A.; All authors have read and agreed to the published version of the manuscript.

**Funding:** This research received no external funding.

**Acknowledgments:** The authors would like to thank David Miles and Andrew Howarth for providing the CASSIOPE telemetry data used for verification in this paper.

**Conflicts of Interest:** The authors declare no conflict of interest.

**Appendix A. Angular Velocity and Bias Estimation**

The Angular velocity is filtered when estimating the bias vector in the $\text{EKF}_\omega$. The system model equations are Equations (3), (A1) and (A2):

$$\boldsymbol{\omega}_{mes} = \boldsymbol{\omega} + \boldsymbol{\beta} + \boldsymbol{\eta}_v \tag{A1}$$

$$\dot{\boldsymbol{\beta}} = \boldsymbol{\eta}_u \tag{A2}$$

where $\boldsymbol{\omega}_{mes} \in \mathbb{R}^3$ is the gyroscope output, $\boldsymbol{\beta} \in \mathbb{R}^3$ is the gyroscope bias vector, $\boldsymbol{\eta}_v \in \mathbb{R}^3$ is the random drift noise and $\boldsymbol{\eta}_u \in \mathbb{R}^3$ is the random walk drift noise. The sate vector is $\mathbf{x} = [\boldsymbol{\omega}^T \ \boldsymbol{\beta}^T]^T$. The Jacobean matrix that will be used for computing the state transition matrix $[\phi]$ is as follows:

$$[F(x)] = \begin{bmatrix} [I]^{-1}([I\bar{\boldsymbol{\omega}}]_x - [\bar{\boldsymbol{\omega}}]_x[I]) & -[\mathbf{1}_{3x3}] \\ [\mathbf{0}_{3x3}] & [\mathbf{0}_{3x3}] \end{bmatrix} \tag{A3}$$

The state transition matrix $[\phi]$ is approximated by $[\phi] \approx [\mathbf{1}_{3x3}] + [F(x)] \, dt$ for small time step $dt$ [63]. The linearized form of the measurements matrix is

$$[H] = \begin{bmatrix} [\mathbf{1}_{3x3}] & [\mathbf{0}_{3x3}] \end{bmatrix} \tag{A4}$$

**Appendix B. Angular Velocity Estimation**

The Angular velocity is filtered and estimated for the real data case where the system model equations are Equation (3). The Jacobean matrix that will be used for computing the state transition matrix $[\phi]$ is as follows:

$$[F(x)] = [I]^{-1}([I\bar{\boldsymbol{\omega}}_{bi}]_x - [\bar{\boldsymbol{\omega}}_{bi}]_x[I]) \tag{A5}$$

The linearized form of the measurements matrix is

$$[H] = [\mathbf{1}_{3x3}] \tag{A6}$$

**Appendix C. Magnetic Field Estimation**

The magnetic field is filtered and estimated for the real data case where the system model equation is Equation (18) and the measurement is the pseudo measurements, see Section 3.2. The Jacobean matrix that will be used for computing the state transition matrix $[\phi]$ is as follows:

$$[F(x)] = [R(\mathbf{q}_{k|k-1})][R(\mathbf{q}_{k-1|k-1}^{-1})] \tag{A7}$$

The linearized form of the measurements matrix is as in Equation (A6) .

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
