# Peer review of "A Recursive Approach for Magnetic Field Estimation in Spacecraft Magnetic Attitude Control"

_aerospace, doi:10.3390/aerospace9120833_

Round 1

Reviewer 1 Report

General comment

The maniscript is well written, the work covers a topic of current interest and the solution proposed is innovative enough for a research paper.

However several aspects shall be improved

- The authors shall clearly quantify (i.e. based on a test case) the improvement resulting from the implementation of the proposed method (i.e. savings in power due to reduced on-off usage of the coils, better pointing accuracy, ...). Data from one (or more) test case(s) could be collected in a table for the sake of compactness.

- The implementation of the method on the on-board computer of a satellite, especially those of micro-/nano-satellites which usually rely on magnetic ADCS, seems computationally expensive. The authors could introduce a comment on this. In particular, could the method be used for real-time attitude determination onboard such a satellite?

- The review on magnetometer-only attitude determination methods is poor and shall be improved.

Please check also the minor issues below

Introduction

- Line 49) "These models, however, are sometimes inaccurate [2,16,17]". These references point out inaccuracies that are relevant only on secular timescales or are of an ephemeral order-of-magnitude for the sake of the process discussed in this work. What is their contribute in here? Please provide a detailed explanation in the manuscript or remove them.

Numerical simulations

- Line 367-370) "A random Gaussian process is..." Please provide the characterizing parameters. The same for noise and similars.

Verification using real data

- Line 479) "...with a duty cycle of δ = 0.7...". Also the period (or the duty cycle value in seconds) shall be reported to provide a complete information.

- Lines 501-504) Please indicate how the covariances matrices were selected or add a reference for it.

Conclusion

- It is stated in the Introduction "The proposed concept is to eliminate the need to turn off the  agnetic rods, in some cycles, to increase the rod’s operation time." and then in the Conclusion "...the proposed algorithm results in shorter maneuver times as well as less power consumption by the magnetic rods. The results also show a less attitude steady state error...". I suggest adding information which provide quantitative evidence of the results declared in the conclusive statement.     

Reviewer 2 Report

The manuscript is interesting and very well written. It is always a pleasure to review material from Authors (which I did a couple of times in different journals). I have only handful minor remarks, and also noticed some typos in the text which are highlighted in the annotated pdf file.

1. Line 125: central principal axes.

2. Eq. 1 and after: Why introduce “frame of interest”? Inertial frame is used in the paper, so just use it in Eq. 1 right off the bat.

3. Figure 5: Something happens up to 2k seconds only. Either xlim to 2k on X axis, or add a close-up of what happens near 10k seconds. Also, ylim to 6.

4. Figure 6, 7: Why plot starts not from “zero”?

5. Line 404: %5 of what?

Round 2

Reviewer 1 Report

Dear authors, thanks for the detailed answers provided. I only have two minor comments:

Point 2: In Section 4 it is mentioned that the computational load (CL) is determined by comparison, with a reference standard algorithm, of the total computational time per maneuver computed using the tic/toc Matlab function. Even though this approach provides a raw evaluation of the CL when run on a PC, it is not decsriptive of the CL of the proposed algorithm when implemented on flight hardware. In fact, after the implementation the algorithm might be even more effective in terms of CL than expected from Matlab. The authors may indicate in the manuscript that the CL could be significantly improved during the implementation.

Point 3; Please correct the typo “ citeSoken20” at line 53: .Please correct the typo “citeSoken20” at line 53.

Author Response

Response to Reviewer 2 Comments

Point 1: In Section 4 it is mentioned that the computational load (CL) is determined by comparison, with a reference standard algorithm, of the total computational time per maneuver computed using the tic/toc Matlab function. Even though this approach provides a raw evaluation of the CL when run on a PC, it is not decsriptive of the CL of the proposed algorithm when implemented on flight hardware. In fact, after the implementation the algorithm might be even more effective in terms of CL than expected from Matlab. The authors may indicate in the manuscript that the CL could be significantly improved during the implementation.

Response 1: The authors would want to express their appreciation to the respected reviewer for his or her recommendation. The text that follows has been added.

It should be noted that, while this approach provides an approximate estimate of the CL when run on the Matlab environment, it does not explain the CL of the suggested technique when run on flying hardware. In reality, the technique may be considerably more effective in terms of CL after implementation than Matlab suggested, since the CL may be substantially improved throughout the implementation.”

Point 2: Please correct the typo “ citeSoken20” at line 53: . Please correct the typo “citeSoken20” at line 53.

Response 2: Corrected.

(please see attachment.)
